# Genomics Applied to the Analysis of Flowering Time, Abiotic Stress Tolerance and Disease Resistance: A Review of What We Have Learned in *Lolium* spp.

**Elisa Pasquali and Gianni Barcaccia \*** 

Laboratory of Genetics and Genomics for Plant Breeding, Department of Agronomy, Food, Natural Resources, Animals and Environment, University of Padova Campus of Agripolis, 35020 Legnaro, Italy; elisa.pasquali.2@phd.unipd.it
**\*** Correspondence: gianni.barcaccia@unipd.it

**Abstract:** Flowering time, abiotic stress tolerance and disease resistance are important agronomic traits of forage species like *Lolium* spp. Understanding the genetic control of these traits is enabled by the combination of genomic tools with conventional breeding techniques. Flowering time in this genus represents a complex trait due to the differences in the primary induction requirements among the species. In total, 36 QTLs (Quantitative Trait Locus) were identified across all seven linkage groups of Italian and perennial ryegrass involved in the flowering pathways, with several putative orthologous/homologous genes that have been characterized in other major crops. From the perspective of climate change, abiotic stress tolerance has become an essential feature; many QTLs that are involved in the control of plant responses have been identified, and transcriptional studies focusing on drought tolerance reported several DEGs (Differentially Expressed Genes) involved in carbon and lipid metabolism and signal transduction. Due to the incidence of microbial diseases, QTLs useful to developing cultivars resistant to bacterial wilt (*Xanthomonas translucens* pv. *graminis*), ryegrass crown rust (*Puccinia coronata* f. sp. *Lolii*) and gray leaf spot (*Magnaporthe grisea/oryzae*) have been mapped in both *L. perenne* and *L. multiflorum* populations. Due to the great importance of *Lolium* species, especially as forage crops, additional information about the three aforementioned agronomic traits is needed.

**Keywords:** *Lolium multiflorum*; *Lolium perenne*; applied genomics; flowering time; abiotic stress; disease resistance

## 1. Introduction

The genus *Lolium*, which includes the two economic species *L. perenne* (perennial ryegrass) and *L. multiflorum* (Italian ryegrass), fulfills most forage needs in Europe, New Zealand and the temperate regions of Japan, Australia, South Africa and South America. Its importance is derived from its high productivity and its high nutritional value as livestock feed, especially in terms of fiber palatability and digestibility. Perennial ryegrass is also widely sown in amenity lawns and as sport turf. Over the years, advantages have also been derived from these grasses from an environmental point of view; Italian ryegrass in particular is used as a soil stabilizer, for example, during the winter to provide a ground cover against soil erosion and depletion [1].

The species belonging to the genus *Lolium* are naturally diploid, with seven pairs of chromosomes, although tetraploid forms have been developed by chromosome doubling to achieve higher biomass production. The genome is relatively large (1C = 2567 Mb for Italian ryegrass and 1C = 2623 Mb for Perennial ryegrass) and, although *L. rigidum*, *L. temulentum*, *L. remotum* and *L. loliacuem* are regarded as self-fertile species [2], the two economic species are natural cross-pollinators, with a high degree of

self-incompatibility regulated by both S and Z loci with many alleles [3]. The Mediterranean basin appears to be the most likely site of origin of this genus based on its shared genomic ancestry with other Eurasian cereal species, such as wheat, rice and barley [4–6].

Perennial ryegrass differs from Italian ryegrass not only in its morphological traits but also in its different vegetative habit, since *L. multiflorum* is a biennial or annual species. To be more precise, the truly annual forms of *L. multiflorum* (Westerwolds ryegrass) can be distinguished by their complete flowering and consequent seed production in the year of sowing. This particular habit is probably the result of selection by grassland farmers in the northern part of the Netherlands (from which the name 'Westerwolds' originates), who in the late 19th century operated repeated and short harvesting cycles in Italian ryegrass fields. On the other hand, biennial forms of Italian ryegrass generally produce very few seed heads in the sowing year, and complete the cycle the following year, remaining leafy during the entire season. Despite its lower persistence and stress tolerance in comparison with *L. perenne*, Italian ryegrass is more productive and is able to provide a faster ground cover, thanks to its timely emergence and seedling vigor [7].

The significant improvement not only in yield but also in the nutritional and agronomic traits of most major crops that has occurred since the beginning of the 21st century gives clear evidence of the value of the use of genomic tools in plant breeding. The combination of conventional breeding techniques and genomic application has proven to be a successful approach. Conventional breeding of forage grasses is mainly based on recurrent phenotypic selection aimed at the production of synthetic populations using, intercrosses or polycrosses of a limited number of parental plants selected for their general combining ability [8]. Nevertheless, the outcrossing reproductive system of *Lolium* spp., coupled with its strong self-incompatibility system, leads to breed cultivars usually characterized by a high level of within-population genetic diversity. Although this confers a great advantage in terms of adaptability to multiple environmental conditions, it may also constitute a difficulty in breeding programs, especially with respect to the fixation of desired traits. Fortunately, molecular techniques provide effective complementary tools that can be used to develop more efficient breeding strategies [9–11]. Because of the significance of the species and the relative lack of information compared with other cereal crops, there is a great interest in developing molecular and conventional strategies for breeding cultivars of ryegrasses, especially strategies for maintaining and stabilizing the genetic variability within populations. There are also differences between *L. perenne* and *L. multiflorum* in terms of the quantity of genomic information available, since the majority of existing studies concern perennial ryegrass. The recent research of Knorst et al. [12], which reported the first assembly of the gene space of Italian ryegrass, undoubtedly provides a useful starting point for efforts to fill the gap between the two related species.

In this review, thanks to the comparison and the syntenic relationship of the genus *Lolium* with other more intensively studied grasses such as rice (*O. sativa*), wheat (*Triticum* spp.) and the model species *Brachypodium distachyon*, three important agronomic aspects have been analyzed in an effort to better understand the mechanisms that control these traits. A cross-sectoral work focused on the flowering time, the abiotic stress tolerance and the disease resistance would report the available information related to a possible genomic application in breeding ryegrasses.

## 2. Flowering Time

Flowering time is one of the most complex and important traits of forage crops, not only due to its agronomic significance but also because of its role in the adaptation of species to specific environmental and geographic zones. As a matter of fact, the transition from the vegetative to the reproductive stage strongly influences feed quality and farm organization. The fiber content of the plant consistently decreases during the reproductive phase, reducing both its digestibility and the amount of metabolizable energy per unit of dry matter. Moreover, several agronomic traits that change together with flowering must be taken into account, such as seasonal herbage yields, required mowing time and farming activities, plant height, leaf length and final seed yield [13,14]. Flowering time is

usually measured by heading date, which is the day on which three inflorescences emerge from the flag leaf sheath of a plant or, in the case of a population, when a fixed threshold of stems (such as 50%) present emerged heads (UPOV, International Union for the Protection of New Varieties of Plants). Heading date is also an important trait that is evaluated in the distinctiveness, uniformity and stability (DUS) test used in the assessment of new varieties.

For most temperate grasses, flowering is the result of the primary induction of low temperature vernalization and short days, followed by higher temperatures and long day conditions that promote heading and anthesis [15]. In the genus *Lolium*, flowering time represents quite a complex topic due to the different growth habits that can be distinguished among the species. The primary induction requirements range from obligatory in *L. perenne* to none in *L. temulentum* and, in summer, annual varieties of *L. multiflorum* (Westerwolds ryegrass). In *L. multiflorum*, it is possible to identify three different types that differ in growth habit: Italian ryegrass, a biennial type with intermediate primary induction requirements, the winter annual strains with similar needs, and the summer annual form Westerwolds ryegrass, which requires neither cold nor short days for anthesis [16,17]. Cooper [16] demonstrated that 50% of biennial Italian ryegrass plants show obligatory cold and short-day requirements for induction, and that the residual 50% exhibit a quantitative response, producing heads in the absence of any inductive treatment. As mentioned above, the geographic origin of the population plays a pivotal role in determining the inductive process required for flowering. A clear example is the Mediterranean cultivar "Vejo", which shows a quantitative long-day response with no cold requirement, in contrast to other cultivars that originated in Continental Europe and need at least 9 weeks of exposure to 6 °C temperatures under short-day conditions for saturation of the primary induction requirement. In the latter case, low-temperature vernalization is the most important factor controlling the passage to the reproductive developmental stage [17].

Due to the lack of information regarding the genetic control of flowering time in *L. multiflorum*, it is reasonable to assume that the biennial and winter types have regulatory mechanisms comparable to those of *L. perenne*, as summarized in Table 1. Overall, 36 QTLs (Quantitative Trait Locus) associated with heading date have been detected in perennial ryegrass in several studies [18–27]. It is essential to underline that, since they were identified in different studies, the correlation between linkage groups and the identity of the marker loci involved can hardly be determined due to the distinct mapping populations and trial locations used, the lack of a reference genome, and the scarcity of common molecular markers to be used as bridges among distinct genetic linkage maps. Nevertheless, the results of these studies confirm the strong genotype-by-environment interaction that always has to be considered in studying heading date. Synteny with the wheat Vrn1 gene was identified on LG 4 [21], conserving the same function in both diploid wheat and perennial ryegrass. The same region on LG 4 may be coincident with a putative casein kinase gene that is involved in the photoperiodic response and has been mapped in the rice genome [26]. The QTL found on LG 7 was thought to be associated with the gene LpCO, which is homologous to Hd1 of rice and CONSTANS of *Arabidopsis* and affects the photoperiodic sensitivity of the flowering process [28,29]. Moreover, a syntenic relationship was found with the Hd3 region identified in rice, which encodes the FLOWERING LOCUS T (FT) ortholog of *Arabidopsis* and controls the induction of the reproductive phase at the meristem [29,30]. Skøt [31] reported the active role of the FT gene in regulation of the flowering response in *L. perenne*. It could be interesting to investigate the same genomic regions in biennial and winter forms of *L. multiflorum*, in order to verify not only the presence but also the structure and the putative functions of the possible homologous genes.

As is also known for well-studied monocotyledonous plants, the genetic control of flowering operates through a very complex molecular network that integrates three different major pathways, namely, the vernalization, photoperiod and autonomous pathways [32]. In perennial ryegrass, the putative ortholog LpFT3 was found on LG 7 [33], and its expression follows the circadian clock. Interestingly, mutations in the promoter region of LpFT3 were strongly associated with changes in flowering time [31]. Although a putative ortholog, LpSOC1 (SUPPRESSOR OF OVEREXPRESSION



OF CONSTANS1), with a not-so-clear function, has been mapped to LG 6 [34], other genes such as MADS-box genes LpMADS1 (identical to VERNALIZATION 1, LpVRN1) and LpMADS2 appear to have a SOC1-like role [35,36].

In more detail, Chouard [37] defined vernalization as "the acquisition or acceleration of the ability to flower by a chilling treatment", and this is confirmed by the requirement of several weeks of low temperature together with short days for primary induction. Thus, the vernalization pathway combines temperature-associated signals with the timing of transition to flowering after a period of cold. In cereal crops, vernalization induces the expression of VRN1 genes, which subsequently control the expression of VRN3, which promotes flowering and in parallel downregulates VRN2 and ODDSOC2, repressing flowering [38]. In *L. perenne*, LpVRN1 has been mapped to LG4 as the putative orthologous sequence of the *T. monococcum* TmVRN1 gene [21]. Moreover, the expression of three MADS-box genes, i.e., LpMADS1, LpMADS2 and LpMADS3, was shown to be upregulated by vernalization [39]. Interestingly, based on the phylogenetic analysis, it is likely that LpMADS1 is identical to the LpVRN1 locus [40], but, as previously mentioned, LpMADS1 appears to have a function more similar to that of *Arabidopsis* SOC1. Due to their similar patterns of protein–protein interaction, a partially redundant role is played by LpMADS2 in vernalization flowering induction. Finally, LpMADS3 seems to maintain an AP1-like function (APETALA-1), promoting flowering and identifying floral meristem [36]. Since the expression levels of two candidate sequences for an *L. perenne* ortholog of TmVRN2 (LpVRN2-2 and LpVRN2-3) were shown not to be associated with cold treatment [41], it could be assumed that a VRN2 ortholog is absent from the perennial ryegrass genome, as confirmed in studies of the model grass species *Brachypodium distachyon* L. [42]. In addition, LpJMJC (a JUMONJI-like gene), LpCOL1 (a member of the CONSTANS-LIKE gene family) and LpOX1 (a putative 2OG-Fe(II) oxygenase) may also be involved in the vernalization response [40].

The photoperiod pathway connects floral induction to variations in day length and interacts with the circadian clock. In perennial ryegrass, as probably in the biennial and winter forms of Italian ryegrass, long-day conditions signal an essential need for floral development, whereas short-day conditions maintain the vegetative stage. Although photoperiod regulation has not been as well studied as the vernalization response, the photoperiod pathways in the model plant *Arabidopsis thaliana*, temperate grasses and cereal crops appear to be well conserved. Two key genes characterize this molecular pathway, namely, GIGANTEA (GI), which acts upstream in tight association with the circadian clock mechanism, and the AtCO (CONSTANS) gene, which directly promotes flowering [42]. In monocots that are able to promote flowering after a specific day length induction, several CO-like genes, such as QTL OsHD1 (Heading Date) in rice [43], TaHD1-like genes in hexaploid wheat that are orthologs of OsHD1 [44], and HvCO1–HvCO9 in barley have been detected. Thanks to Martin et al. [28], the LpCO gene has also been identified in LG 7 in the perennial ryegrass genome, based on its sequence similarity to AtCO and OsHD1. Interestingly, LpCO has an organization similar to that of AtCO, with an intron characterized by the zinc finger domain, and the promoter region sequence appears to be highly related to the corresponding regions of orthologous OsHD1 and HvCO. As a confirmation of the influence of circadian clock mechanisms on the expression of CO-like genes, LpCO transcriptional levels showed diurnal oscillations, with a significant increase under long-day conditions. The studies of Armstead et al. [29] report the identity of LpCO with the LpHD1 gene. The LpCOL1 gene needs further investigation to allow us to better understand its function in this process. The homologous LpGI gene for ryegrasses has been identified on LG 3 in a syntenic region to which both *Brachypodium* and rice GI genes have been mapped, and which shares a conserved structure with them. Moreover, the expression of LpGI is modulated by the circadian rhythm, responding to day length as is also observed in other grasses [45]. Perennial ryegrass harbors only one copy of the GI gene, as was also confirmed for the model temperate grass *Brachypodium*, with no evidence of the presence of extra GI copies [42].

Other genes involved in the circadian clock mechanism have been identified in the ryegrass genome, namely, Lpck2a-1 and Lpck2a-2, which encode two putative a-subunits of casein protein

kinase 2 (CK2), and their expression shows a strict association with variation in heading date and winter survival [46]. Additional roles in the oscillation of the circadian clock are played by LpLHY and LpTOC1, which have been mapped to LG 7 and LG 6, respectively, as the homolog sequences of LATE ELONGATED HYPOCOTYL (LHY) and TIMING OF CAB EXPRESSION1 (TOC1) in *Arabidopsis* and *Brachypodium* [42,47].

The third important pathway involved in floral development is the so-called "autonomous" pathway; it is named for its role in the flowering of late-flowering mutants irrespective of day length. All genes belonging to this pathway function as repressors of FLC (FLOWER LOCUS C) expression; since FLC is the key flowering repressor in *Arabidopsis*, a delay in flowering results from a high expression of FLC in mutants. In other words, genes of the autonomous pathway generally promote floral development indirectly by repressing the floral repressor FLC. Further studies are necessary in order to better understand this mechanism in grasses, especially because no FLC ortholog has yet been detected; however, two genes of the autonomous pathway, LpFCA and LpFY, were identified in ryegrass based on their similar structures and conserved DNA binding sites [48]. The major difficulty encountered in investigating this pathway in ryegrasses is that the corresponding VRN2 gene, which represents the ortholog of FLC in grasses [49], as previously reported, has not yet been identified; thus, the functions and targets of autonomous pathway genes in ryegrasses are not so clear.

Focusing on the summer annual form Westerwolds ryegrass, which requires neither cold nor short days for floral induction, it can be assumed that mutations on the first intron of the LpVRN1 gene can produce changes in the vernalization response [50]. In fact, a deletion within this intronic region has been found to be responsible for the lack of vernalization requirements in the Vejo cultivar [51]. Confirming and similar evidence has also been reported for the spring growth habit of varieties of wheat and barley, which is determined by allelic variation at the VRN1 and/or VRN2 loci [52]. Therefore, specific studies are needed to clarify and possibly confirm this hypothesis in *L. multiflorum* summer accessions.

**Table 1.** Key genes involved in the control of flowering time.

| Gene | Position | Putative Ortholog/Homolog | Pathway | References |
|---|---|---|---|---|
| LpVRN1 (LpMADS1) | LG4 | TmVrn1 (AtSOC1) | Vernalization | [21,34,36,39,40,53] |
| LpSOC1 | LG6 | AtSOC1 | | [34] |
| LpMADS2 | | LmMADS2 | Vernalization | [35,39] |
| LpMADS3 | | AtAPI1 | Vernalization | [35,36,39] |
| LpLMLC | | JUMONJI-like | Vernalization | [40] |
| LpOX1 | | 2OG-Fe(II) oxygenase | Vernalization | [40] |
| LpCOL1 | LG6 | AtCO, OsHd1 | Vernalization, Photoperiod | [40] |
| LpCO (LpHD1) | LG7 | AtCO, OsHD1, HvCO | Photoperiod | [28,29,34,53] |
| LpGI | LG3 | OsGI, AtGI, HvGI, Brachypodium GI | Photoperiod, circadian clock | [34,42,45] |
| Lpck2a-1 | LG4 | Casein kinase 2a | Circadian clock | [46] |
| Lpck2a-2 | LG2 | Casein kinase 2a | Circadian clock | [46] |
| LpLHY | LG6 | AtLHY, Brachypodium LHY | Circadian clock | [42,47] |
| LpTOC1 | LG7 | AtTOC1 | Circadian clock | [42,47] |
| LpFCA | | AtFCA | Autonomous | [48] |
| LpFY | | AtFY | Autonomous | [48] |

## 3. Abiotic Stress Control

Cold, heat, drought and salt can decrease crop productivity by altering the normal physiological, molecular, biochemical and morphological equilibrium. It is expected that abiotic stresses will play an

increasingly key role in crop management in the future due to global climate change [54,55]. Salt stress harms cells by disrupting their ionic and osmotic equilibrium [56], and excessive heat provokes the overproduction of free radicals, producing oxidative stress and placing irrevocable limitations on plant growth [57]. Cold stress constrains many physiological processes by inducing the production of reactive oxygen species (ROS), which limit photochemical efficiency and consequently hinder photosynthesis [58].

Genetic improvement to develop plants that are more tolerant to abiotic stresses is challenging due to the complexity of the processes involved in abiotic stress responses. Cold acclimation and freezing tolerance are resulting responses to the complex interaction between changed environmental conditions, such as low temperature and light exposure, and consequent photosystem II excitation pressure.

Drought tolerance is also the result of combining traits: it involves plant survival, the ability to grow quickly after rewatering and very slowly under drought conditions, and the ability to adjust leaf transpiration to root water absorption.

The molecular basis of these tolerances can be better explained through the possibility that candidate genes cosegregate with QTLs associated with stress tolerance. A clear example of this is the cosegregation of two dehydrin loci (Dhn1/Dhn2 and Dhn/9) with cold and salt tolerance QTLs, and with an abscisic acid (ABA) QTL located in the same region of chromosome group 5 in the Triticeae, as for the vernalization and frost tolerance loci Vrn-1 and Fr1 [59]. Dehydrins, indeed, play a key role in abiotic stress responses, especially in dehydration tolerance, due to their ability to stabilize membrane function by acting as chaperones to inhibit the aggregation and/or inactivation of proteins under dehydration conditions [60]. Other dehydrin genes (Dhn3, Dhn4, Dhn5, Dhn7 and Dhn/8) cosegregate with drought tolerance QTLs on chromosome 6. Therefore, it is possible to assume that the quantitative control of stress tolerance could be managed by a regulatory gene that is able to regulate the expression of multiple stress-related genes. This is what occurs in wheat in the expression of the cold-induced protein cor14b, which is controlled by Rcg1 and Rcg2 loci in the Vrn-1/Fr1 region on chromosome 5A [61].

As confirmed by the genomic map constructed by Cattivelli et al. [62], in Triticeae, group 5 chromosomes represent the most important part of the genome involved in stress tolerance; this part of the genome includes major loci and QTLs associated with heading date, as well as frost and salt tolerance. Another significantly conserved region related to drought tolerance has been identified on chromosome 7. The presence of QTLs involved in multiple stress responses implies that plants possess common mechanisms that they can implement against different stresses, and that some plant genes may be clustered on the basis of how they contribute to specific tolerance processes. Other key molecules that provide good examples of this are the proteins encoded by the LEA (Late-Embryogenesis-Abundant) gene family, which is induced under dehydration, cold conditions, after strong ABA activity and during osmotic stress, and is fundamental in several abiotic stress conditions. LEA proteins, in fact, are extremely hydrophilic molecules acting in cells experiencing changes in water status due to their high content of water-interacting residues, preserving macromolecules and cellular structures [63,64]. Yu et al. [65] reported a putative LpLEA3 in *L. perenne* that may facilitate drought tolerance: in particular, two amino acid substitutions in the product of this gene have been identified that may enhance the hydrophilic nature of this peptide, and thereby increase cellular water retention fundamental to drought tolerance. Moreover, the homologous cold-stimulated Lcs19 gene identified in *L. multiflorum* [66] through a BLAST analysis of the NCBI database shows a significant degree of similarity (up to 75%) with cold-responsive proteins found in other cereals, such as *T. aestivum*, *A. speltoides* and *H. vulgare*.

Taking into account the findings regarding abiotic stress responses in the *Lolium–Festuca* complex, Alm et al. [67] produced a QTL map for frost tolerance, winter survival and drought tolerance in meadow fescue (*Festuca pratensis* Huds). Two major QTLs for freezing tolerance and four QTLs for winter survival were identified on LG 4, and, thanks to the availability of heterologous wheat anchor probes, it is possible to consider the Frf4-1 locus as orthologous to the frost tolerance loci Fr1 and Fr2 in wheat, whereas the QTL for winter survival mapped on *Festuca* LG 1 is orthologous to a water-soluble

carbohydrate QTL in *Lolium*. Four putative QTLs associated with moderate drought tolerance (two months without water) were also detected; among these, the QDtm1F-1 locus on LG 1 and the QDtm4F locus on LG 4 showed major effects, and the QTL mapped to LG 6 may be orthologous to an ABA locus in wheat. Six additional QTLs have been identified in plants subjected to severe drought conditions (five months without water), and among these the QDts5F-1 locus mapped on LG 5 represents a major QTL for severe drought tolerance. Interestingly, meadow fescue LG 3 can constitute a major source of several drought tolerance genes due to the presence of QTL QDts3 along its entire length. *Festuca* LG 3 is syntenic with rice chromosome 1 along its entire length, including QTLs involved in rooting (a drought-adaptive trait) and osmotic adjustment (a drought-induced trait) [68]. Therefore, QDts3 may be associated with root development under dehydration conditions. Sequences of *Festuca* LG 3 have also been reported to be associated with salinity tolerance in some *Festuca* × *Lolium* hybrids [69,70]. In fact, introgression of *F. pratensis* genes in *Lolium* genomes has often represented a great way to also improve freezing and drought tolerance, developing hybrid plants [71–73].

Little information is available on the QTL mapping of drought tolerance traits in *Lolium* species. It is known from the study of Turner et al. [74] that a possible cluster is located on LG 1 and LG 5; compared with the orthologs observed in meadow fescue, thanks to the use of common markers, it is likely that the QTLs might well colocate.

More recent studies were able to analyze the transcriptional response and the consequent expression of some genes that are differentially expressed in plants grown under various conditions. Lee et al. [75] observed several differentially expressed genes (DEGs) in the leaf tissue of Italian ryegrass seedlings exposed to multiple abiotic stresses, in particular cold, salt and heat (Table 2). After validation of the sequences of the DEGs by comparison with corresponding homologs in the NCBI database, it is possible to list some of the putative genes involved in abiotic stress mechanisms; these genes are generally implicated in photosynthesis, vegetative development, protein synthesis and abiotic stress tolerance. Among them, a light-harvesting chlorophyll a/b binding protein (LHCB) has been identified; in the literature, LHCB is reported as a key apoprotein of the light-harvesting complex of photosystem II, acting in guard cell signaling. Due to its upregulation in cold conditions and according to recent findings in barley [76], it is likely that this protein plays a crucial role in photosynthesis and leaf growth under low-temperature conditions. Another cold-induced gene found in Italian ryegrass shows homology with the elongation factor 1 alpha (EF-1 alpha) of *Deschampsia antarctica*, a native Antarctic grass that is able to survive in extreme environmental conditions. In crop species, EF-1 alpha seems to be associated with the higher levels of protein synthesis in developing leaves in tomato plants [77]. In addition, the enzyme glyceraldehyde-3-phosphate dehydrogenase (GAPDH) exhibited differential expression after both cold and salt induction. This enzyme is a pivotal molecule in glycolysis, and, as confirmed in previous studies in wheat, it is associated with major abiotic stress tolerance [78]. Thus, the upregulation of the GAPDH gene suggests its importance in tolerance mechanisms in ryegrasses. Moreover, two other DEGs induced by salt and heat were detected; these DEGs are involved in leaf elongation and internode growth during the vegetative stage, and were identified due to their homology with the alpha-galactosidase b ($\alpha$-Gal b) gene in *Aegilops tauschii* subsp. *tauschii* and the translation initiation factor Shortened Uppermost Internode1 (SUI1) in *Brachypodium distachyon* [60,79].

**Table 2.** Differentially expressed genes (DEG) putatively involved in abiotic stress tolerance in *L. multiflorum*.

| DEG | Biological Pathway | Abiotic Stress | Similar Findings in Other Species |
|---|---|---|---|
| LHCB | Cell signaling | Cold | Barley [76] |
| EF-1 $\alpha$ | Leaf development | Cold | Tomato [77] |
| GAPDH | Glycolysis | Cold and salt | Wheat [78] |
| $\alpha$-Gal b | Leaf development | Salt and heat | *A. tauschii* [79] |
| SUI | Leaf development | Salt and heat | *B. distachyon* [60] |
| LDH | Glycolysis | Drought | Rice [80] |
| GAPCs | Glycolysis | Drought | Arabidopsis [81] |
| TPP | Starch and sucrose metabolism | Drought | Tobacco [82] |
| PGM | Starch and sucrose metabolism | Drought | Wheat and *Arabidopsis* [83,84] |
| GSR | Glycerophospholipid metabolism | Drought | Corn [85] |
| GGT | Glycerophospholipid metabolism | Drought | Corn [86] |
| AP | Glycerophospholipid metabolism and ascorbate pathway | Drought | Grapevine [87] |
| PIase C | Signal transduction pathway | Drought | Rapeseed [88] |
| NPR1 | Signal transduction pathway | Drought | Rapeseed and *Arabidopsis* [89,90] |

Although there is little information regarding drought-related genes in Italian ryegrass, thanks to the transcriptional analysis of Pan et al. [91], it is possible to better understand the molecular mechanisms of drought tolerance in this important forage species (Table 2). The genes encoding drought-related regulatory proteins play pivotal roles in carbon and lipid metabolism and in signal transduction. As alterations in glycolysis and gluconeogenesis form the basis of plant adaptations to abiotic stress, drought results in modified sucrose and amino acid content [92]. Several genes that encode enzymes involved in glycolysis and gluconeogenesis showed upregulation in tolerant plants, including the gene encoding the hydrogen transfer enzyme L-lactate dehydrogenase (LDH), which catalyzes the reversible conversion of pyruvate to lactate, and the genes encoding glyceraldehyde-3-phosphate dehydrogenases (GAPCs), which are involved in glycolysis pathways; these observations are consistent with observations reported in *Arabidopsis*. The latter enzyme may represent the connector between membrane lipid-based signaling, energy carbon metabolism and growth control in plant responses to ROS and drought stress [81]. The important connection with a limited photosynthetic process due to drought stress can be partially explained by the possible alleviating function of phosphoenolpyruvate carboxylase (PEPC), which is able to balance carbon and nitrogen metabolism [65,93]. Interestingly, a strong relationship between the expression of genes related to starch and sucrose metabolism and drought stress has been observed, suggesting that Italian ryegrass seedlings expend considerable resources in the production of direct and immediate energy sources. Clear examples are provided by the upregulation of genes encoding trehalose-6-phosphate phosphatase (TPP) and phosphoglucomutase (PGM) reported in drought-resistant lines; these enzymes are able to modify carbohydrate allocation and metabolism [82], and control sucrose biosynthesis and the distribution of photosynthate between sucrose and starch [94,95], as found also in *Arabidopsis* [83] and wheat [85]. Moreover, substantial alterations in lipid metabolism have also been observed under drought conditions. Pan et al. [91] recently identified five upregulated genes in drought-resistant Italian ryegrass cultivars. The upregulation of genes encoding two enzymes that are important in glycerophospholipid metabolism, glutathione reductase (GSR) and gamma-glutamyl transpeptidase (GGT), may contribute to enhanced drought tolerance. L-ascorbate peroxidase (AP) can also be considered a primary enzyme that is able to increase

drought tolerance by controlling glycerophospholipid metabolism and the ascorbate pathway, based on its reported upregulation in drought-tolerant plants and on its pivotal role in the ascorbate–glutathione cycle, which represents a principal step in free radical detoxification [87]. Moreover, two other overexpressed enzymes involved in the same detoxification process, namely, a-6-phosphogluconate dehydrogenase (6-PGD) and glucose-6-phosphate 1-dehydrogenase (G6PDH), can be considered as potential breeding targets. These enzymes also act as major sources of NADPH in the cytoplasm of plant cells. One of the most important parts of the plant response to abiotic stress is signal transduction, including the modification, delivery and assembly of signaling elements [96]. To give some examples, phosphatidylinositol-specific phospholipase C (PIase C) appears to be upregulated in tolerant lines of Italian ryegrass, as previously confirmed in studies of canola by Georges et al. [88]. This enzyme, indeed, plays a key role in the phosphatidylinositol-specific signal transduction pathway, and can therefore strongly affect drought tolerance through the regulation of signal transduction. A significant number of the DEGs reported in the same study encode regulatory molecules, such as nonexpressorofpathogenesis_relatedgenes1 (NPR1), which has been implicated in the suppression of the jasmonic acid (JA) pathway mediated by the salicylic acid (SA) signaling pathway [97]. Previous studies showed its importance in regulating basal and systemic acquired tolerance in several species [98], suggesting a key function also in drought tolerance in *L. multiflorum*.

As a confirmation of previous research, Pan et al. [99] gave a detailed overview of the differentially accumulated proteins synthetized in response to short-term drought, comparing the proteomic profiles of resistant and susceptible plants. The results of this analysis suggest that drought-responsive proteins are involved in carbohydrate and amino acid metabolism, the synthesis of secondary metabolites, and signal transduction pathways.

## 4. Disease Resistance

### 4.1. Bacterial Wilt—Xanthomonas translucens pv. graminis

The most important bacterial disease of forage crops is bacterial wilt, and, among forage grasses, Italian ryegrass is considered the most susceptible to this disease. The causative pathogen is the Gram-negative bacterium *Xanthomonas translucens* pv. *graminis*; it infects the host by entering through leaf stomata and epidermal wounds and multiplying in the xylem, and the infection is promoted by mowing and can be propagated by the use of contaminated mowing equipment. This foliar disease causes severe damage, especially in terms of biomass yield losses, which range between 20% in field lawns and up to 80% with direct leaf inoculation under experimental conditions [100,101].

Since the chemical control of bacterial diseases is difficult, breeding for more tolerant cultivars appears to be the most ecologically and economically viable way to achieve plant protection against *X. translucens*.

Resistance genes are often arranged in clusters, as has been demonstrated for *Xanthomonas* spp. resistance genes in many crop species, including rice (*Oryza sativa* L.) [102], pepper (*Capsicum* spp.) and tomato (*Solanum lycopersicum* L.) [103].

In *L. multiflorum*, while investigating a biparental mapping population, Studer et al. [104] identified a major QTL on LG 4 that explained up to 84% of the observed phenotypic variation. Interestingly, this QTL was stable across each type of experiment under both greenhouse and field conditions, and it is closely linked to the nearest genetic AFLP marker (P38M50_252) at a distance of less than 1.3 cm. This finding could be important considering that the major broad-spectrum resistance gene Xa21 in rice is located on chromosome 11, which was demonstrated to be syntenic to the *Festuca/Lolium* LG 4 [105]. Moreover, in the same study, other minor QTLs were identified on LG 1, LG 5 and LG 6, explaining 3%, 7.4% and 7.3%, respectively, of the observed phenotypic variation. This result confirms findings in other crop species, in which resistance to *Xanthomonas* spp. was shown to be mainly associated with a few major genes or QTLs and other minor loci [106,107].

Since the use of molecular markers in breeding programs is increasingly common, a set of 21 SNPs (Single Nucleotide Polymorphisms) significantly associated with bacterial wilt resistance was recently developed by Knorst [108]. Overall, these SNPs explain up to 41% of the observed variation. In this way, several candidate genes that may be involved in resistance mechanisms were identified, although a level of genetic resolution sufficient to identify the causative genes conferring resistance has not been achieved. Among them, there is the gene that encodes CRINKLY4 (serine/threonine protein kinase-like protein CCR4), which might be involved in the detection of the bacterium and the consequent initiation of pathogen-associated molecular pattern (PAMP)-triggered immunity (PTI), as found in rapeseed (*Brassica napus*), operating as a pathogen recognition receptor (PRR), leading to PTI after pathogen detection [109].

Another candidate gene for bacterial wilt resistance detected in *L. multiflorum* by Knorst [108] encodes a noduline MtN3 family protein (also termed SWEET17 in *Arabidopsis*). Interestingly, its homolog in rice, SWEET13, was demonstrated to encode a protein (known as xa25) which confers resistance against *X. oryzae* pv. *orizae* [110]. SWEET transporters appear to be a target of vascular pathogens due to their exploitation of sugars transported into the apoplast as an energy source. Therefore, a mutation in the sugar transporter may lead to plant resistance as a result of the limited accessibility of energy.

Another scaffold contains the sequence encoding the protein BAK1 (Brassinosteroid Insensitive 1-associated kinase-interacting receptor-like kinase 1), which is involved in PTI and forms a heterodimer with FLS2 (flagellin sensing 2) after recognition of the bacterial-derived PAMP flg22. Linked to this, a transthyretin-like protein was also detected as a potential substrate for BRI1 (Brassinosteroid Insensitive 1). In this way, it is likely that *L. multiflorum* possesses a mechanism for the recognition of *X. translucens* pv. *graminis* similar to the one known in Arabidopsis for the detection of a conserved region of the bacterial flagellum, flg22 [111]. However, although this pathogen seems to lack the sequences that encode flagellar proteins, a trace of this mechanism may still be present as part of a cluster of resistance genes that are useful against other pathogens.

## 4.2. Ryegrass Crown Rust—*Puccinia coronata f. sp. Lolii*

Ryegrass crown rust, which is caused by the Basidiomycete fungus *Puccinia coronata* f. sp. *Lolii*, causes severe foliar damage to ryegrasses, causing not only great quantitative losses in both dry and green matter yield but also reduced forage quality. As observed, after rust sporulation, the leaves present several breaks; these breaks increase transpiration water loss and reduce general plant vigor due also to the depletion of leaf-blade water-soluble carbohydrate content [112,113]. Because of this decay, a reduction in the thousand-seed weight can be significantly important in ryegrasses cultivated for seed production. In terms of quantitative losses after a crown rust infection, decreases of 56% in dry matter and of as much as 94% in green matter have been reported [114]. As previously anticipated, clear consequences are present also in the nutritional characteristics of the forage, which result in a lowered production of milk due to reduced digestibility and palatability and a general rejection by cows of highly infected grasses [115].

Breeding for enhanced tolerance and resistance in ryegrass cultivars appears to be the best approach to the management of this problem, considering the high economic and environmental costs associated with chemical and agronomic control methods. To achieve a complete form of resistance, the presence of nucleotide binding site–leucine-rich repeat (NBS-LRR) genes is fundamental [116], since these genes encode receptor proteins that are able to identify elicitor molecules specific to pathogenic strains. This type of mechanism is called "qualitative" resistance due to the ability of a single gene to make a susceptible individual into a resistant plant, inducing a hypersensitive response in infected cells. Several studies of plant resistance have distinguished a different type of resistance, defined as "slow-rusting" or adult plant resistance, that confers major resilience against pathogen mutations through its structure, thereby avoiding loss of resistance [117]. During infection, the synthesis of several classes of antimicrobial molecules is induced, such as reactive oxygen species (ROS), phytoalexins,

phytoanticipins and pathogenesis-related (PR) proteins. Therefore, the plant implements a downstream response to a pathogen-specific attack in an attempt to inhibit colonization by the rust [118].

In Italian and perennial ryegrasses, despite what was found for other diseases, crown rust resistance appears to be controlled by cytoplasmic mechanisms [119] as well as by both quantitative and qualitative nuclear genetic mechanisms [120]. These different resistance mechanisms, involving multiple major and minor genes, result in more durable disease control compared to mechanisms that involve the regulation of a single major gene. Ryegrass crown rust resistance is mainly polygenic or oligogenic, since most genes confer partial resistance with a certain degree of additivity. It should be recalled that crown rust resistance genes have been introgressed into the *L. multiflorum* genome from the closely related species meadow fescue (*Festuca pratensis*) using a backcrossing program; these genes were later identified in the distal part of the short arm of LG 5 [121].

Traditional genetic improvement programs for increasing crown rust resistance have been based on recurrent phenotypic selection within natural and breeding populations. Thanks to the development of marker-assisted breeding (MAB) techniques, especially those based on SNP and SSR markers, it was possible to identify and evaluate the colocation of potential functional polymorphisms and trait-specific QTLs.

Through the use of genomic tools, resistance-specific QTLs were identified in perennial ryegrass (the majority of studies), Italian ryegrass and interspecific hybrid mapping populations.

In *L. perenne*, QTLs for crown resistance have been found in all seven linkage groups: Dumsday [122] and Muylle [123] identified four major genes that show a large phenotypic effect against an Australian pathotype of the fungus (gene LpPc1) and against European pathogen strains (genes LpPc2, LpPc3 and LpPc4). These genes were later mapped to LG1 (LpPc4 and LpPc2) and LG2 (LpPc3 and LpPc1). Interestingly, these linkage groups are closely related to groups A and B of oat (*Avena sativa*) due to their syntenic relations, and they also present high homology to chromosome 1 of Triticae [124]; on both of these regions, indeed, leaf rust resistance genes have been identified [125]. Several minor QTLs have been identified in various mapping populations.

QTL mapping analysis has also been performed in European and Japanese Italian ryegrass populations. In both greenhouse-controlled and field experiments, two major QTLs are able to explain up to 56% of the phenotypic variation, together with a minor QTL found in some specific locations under some conditions [126], in accordance with what has been reported for *A. sativa*. Basic information on these markers is reported in Table 3.

**Table 3.** QTLs (Quantitative Traits Locus) detected by Studer et al. [126] for ryegrass crown rust in *L. multiflorum*.

| QTL | LG | Closest Marker | Variance Explained (%) |
|-----|-----|-----|-----|
| QTL 1 | 1 | SSR NFFA012 | 56% |
| QTL 2 | 2 | AFLP E35M50_2002 | 35% |
| QTL 3 | 3 | LPSSRK03G05 | 11–13% |

Although the explicit difficulty of determining a correlation between the QTLs detected in the absence of a reference genome and common markers at the time when experiments were taken, it is important to point out the strong interaction between resistance QTLs and environment, since a resistant cultivar can become susceptible if cultivated in a different geographical and ecological environment, due to the developed resistance against specific pathotypes of the fungus.

Various QTLs for ryegrass crown rust were also identified by Sim et al. [127] when investigating an interspecific three-generation Italian ryegrass × perennial ryegrass cross population. After two-year-long trials, they detected four resistance QTLs on LG 2, LG 3 and LG 7, and a small one on LG 6 (Table 4).

**Table 4.** QTLs detected by Sim et al. [127] for ryegrass crown rust in an Italian × perennial ryegrass cross-population.

| QTL | LG | Closest Marker | Variance Explained (%) |
|------|------|------|------|
| QTL 1 | 2 | RFLP BCD1184 | 8–15% |
| QTL 2 | 7 | RFLP BCD782 | 12–19% |
| QTL 3 | 3 | RFLP RZ444 | 10% |
| QTL 4 | 6 | RZ273 | 8–10% |

The QTLs on LG2 and LG7 showed a stable effect under diverse environmental conditions, and it is possible to assume that the LG 2-located QTL corresponds to the LpPc1 locus based on the association of its RFLP markers. However, it cannot be excluded that it is LpPc3, due to the possibility of over- or underestimating map distance in different populations. The minor QTL detected on LG6 was found only in an experimental plot, and can therefore be assumed to be a race-specific QTL.

Both macrosyntenic and microsyntenic relationships link forage species such as ryegrasses to the most studied closely related cereals. Thanks to this downstream knowledge and to the innovative use of genomic tools, it was possible to identify candidate genes for major and minor genes involved in resistance mechanisms in ryegrasses. For Italian ryegrass, 62 distinct R (resistance) gene sequences containing continuous open reading frames (ORFs) have been characterized as putative functional R genes [128]. According to the genomic results of this work, the evolutionary affinities in the sequence structures of resistance genes between the genus *Lolium* and other Poeae grass species have been confirmed. A clear example of this relationship is reported by Jones et al. [124], who compared the locations of several candidate R genes of perennial ryegrass with those of other Poaceae; among these was the leaf rust receptor kinase gene TaLrk10, located within the Lr10 locus in hexaploid wheat [129,130] and able to confer resistance to leaf rust in certain wheat cultivars. Indeed, the putative orthologus of TaLrk10 mapped on perennial ryegrass LG1 shows a conserved syntenic location through the alignment of LG1 of Italian ryegrass, LG4-12 of hexaploid oat (*Avena sativa* L.), chromosome 1AS of wheat (*Triticum aestivum* L.), chromosome 1HS of barley (*Hordeum vulgare* L.) and chromosome 1RS of rye (*Secale cereale* L.) [131,132].

Moreover, 12 perennial ryegrass candidate DR (Defense Response) gene loci, which were identified due to their high sequence similarity to putative orthologs in other cereals, have been identified as coincident with crown rust resistance QTLs in several mapping populations of *Lolium*, based on their common genetic marker positions [133].

In subsequent investigations, coincident QTLs for both lignin biosynthesis and crown rust resistance were detected on LG3 and LG7, indicating a possible positive correlation between these two traits, as was found in other forage grasses such as smooth bromegrass (*Bromus inermis* Leyss.) by Delgado et al. [134]. According to this, cell-wall lignification may be positively involved in the mechanism of resistance to *P. coronata*.

Other DR gene clusters were mapped on perennial ryegrass LG 5 and on LG 6. It is worth mentioning that DR genes encoding glucanases and catalases colocalized with these crown rust QTLs in various mapping populations [133]. For example, both LpGLUCk SNP and two crown rust resistance QTLs, which are able to explain 7% and 16%, respectively, of the observed phenotypic variance of the mapping population, have been mapped on two distinct regions of LG 5 [133,135]. Moreover, a cluster of DR genes, including representatives of defensins, chitinases and thaumatin-like genes, has been detected in a similar chromosomal position. Therefore, a reliable assumption is that a large cluster of DR genes located on narrow as well as specific regions of the LG 5 of *L. perenne* could be implicated in the molecular response against crown rust.

Other crown rust QTLs have been located in a region on LG 2 of *L. perenne* that shows a high level of conserved synteny with LG B of hexaploid oat, which contains major QTLs for resistance to *P. coronata* f. sp. *avenae*. Due to the presence of a cluster of R genes in this region, it is likely that the

identified R genes constitute the most plausible candidates for the LpPc1 locus [136], although further experiments are necessary to establish this functional relationship.

Considering the major QTLs for crown rust resistance, LpPc2 and LpPc4, which have been mapped on LG 1 in three different trait-specific mapping populations [123,126,135], it is possible to assume that they may be associated with European pathotypes of crown rust, due to the geographical origin of the parental genotypes and the trial location.

Another important role in response to crown rust seems to be played by peroxidase (class III) enzymatic activity, as demonstrated in wheat [137]. The LpPERa gene locus is quite certainly coincident with a crown rust resistance QTL identified on LG 2 [127,136], to which the QTL LpPc3 has also been mapped [123], but due to the small number of common genetic markers, a reliable comparison appears difficult [133].

### 4.3. Gray Leaf Spot (GLS)—Magnaporthe Grisea/Oryzae (Anamorph Pyricularia)

The first signs of gray leaf spot disease are small brown spots on leaves and stems that develop in water-soaked spots, followed by circular or oval lesions with gray midpoints and dark-brown margins. In cases of heavy infection, under warm and humid conditions and in particularly susceptible genotypes, the infected leaves die, and whole seedlings may be killed in a few days [138].

The causal pathogen, the ascomycete fungus, also affects a very wide host range that includes rice, in which it induces rice blast disease, as well as other grasses that are susceptible to foliar disease, such as wheat and barley on which it causes blast, and other turf and forage grasses such as tall fescue (*Festuca arundincea* Schreb.), on which it causes gray leaf spot.

Because resistance is controlled by a few major genes, one of the most important points to consider in developing GLS-resistant cultivars such as those found in rice is the breakdown of resistance [139,140]. In fact, many resistant cultivars can become susceptible under field conditions as a consequence of the high genetic variability of the fungus and the narrow resistance specificity of the host. In contrast, it was found that GLS resistance controlled both by major and minor genes may be more durable, with plants maintaining partial resistance potentially for years under pathogenic pressure.

From the perspective of the development of GLS resistance, two major QTLs, LmPi1 and LmPi2, were identified in *L. multiflorum* [138]. Based on a BLASTX analysis, the two allelic forms of LmPi1 showed high similarity to parts of the HvAS1 and HvAS2 barley genes encoding asparagine synthetase. Miura et al. [141] showed that this gene leads to moderate resistance, in which infected plants are characterized by spindle-shaped lesions that are sensitive to environmental factors. Selection for individuals who are homozygous at the associated marker locus p56 could be a functional way to enhance GLS resistance in a marker-assisted breeding program. The second major QTL, LmPi2, was identified in a single-cross-derived F1 population of Italian ryegrass, and was demonstrated to have high broad-sense heritability (66.5%) and to explain most of the total phenotypic variance (69.5%). Since it has been reported that younger ryegrass leaves are the most susceptible parts of seedlings, the results of the study of Takahashi [138] suggest that the LmPi2 locus is a great source of resistance, as it is functional at various leaf ages.

Four other minor QTLs, found on other linkage groups by Curley et al. [142], are summarized in Table 5.

**Table 5.** Additional QTLs for gray leaf spot resistance [142].

| QTL | LG | Closest Marker | Variance Explained (%) |
|-----|----|----|----|
| QTL 1 | 2 | AFLP A-E33M62109 | 5–8% |
| QTL 2 | 3 | RFLP CDO460 | 20–37% |
| QTL 3 | 4 | E3.650 | 4–10% |
| QTL 4 | 6 | C19.390 | 6–10% |

## 5. General Discussion and Concluding Remarks

Agronomically important traits, such as flowering time, abiotic stress tolerance and disease resistance, have to be considered as primary features of a crop, especially when choosing a cultivar that is able to adapt to specific environmental conditions and when managing crops throughout an entire agricultural season. In an ever-changing scenario, especially from the climatic point of view, adaptation and reasoned agronomic management are no longer secondary aspects. In addition, the possibility of applying genomics to plant breeding has now become a precious reality, able to assist in prediction, monitoring and targeted modifications.

An amount of information similar to that available in the scientific literature for the major monocots, such as wheat, rice, barley and maize, is still far from having been obtained for *Lolium* spp., especially in the area of genetics and genomics. Moreover, the disparity between perennial ryegrass and Italian ryegrass in the available genetic resources makes future studies more challenging. A clear example is the great lack of information about the control of flowering times in *L. multiflorum*, which shows an extremely complex response, with different inductive requirements and consequent contrasting flowering habits.

Despite the fact that this missing knowledge represents a significant limitation in ryegrass research, it makes the comparative genomics a fundamental tool, as demonstrated by countless studies including those of Chen et al. [143], Asp et al. [50], Pan et al. [99] and Sim et al. [127]. Because the genus *Lolium* belongs to the tribe Poeae in the Poaceae grass family, which includes major crop species, its syntenic relations have also brought important developments, such as the synteny-based draft genome of *L. perenne* provided by Byrne et al. [144]. On the other hand, the recent work of Knorst et al. [108] offers an important starting point for *L. multiflorum* research, and is indispensable for developing efficient genomic-assisted breeding techniques.

Since several hypotheses have been formulated about putative key genes identified as involved or differentially expressed in all the examined traits, it will be necessary to clarify their functions and structures, as in the case of mutations in the first intron of the LpVRN1 gene, which may be responsible for the different vernalization responses of *L. multiflorum* summer accessions. Moreover, from this review, the importance of producing comparable data that make possible the efficient and necessary collaboration between research groups is clear; for example, sharing the same reference mapping population with common molecular markers would greatly advance progress, especially in QTLs analysis, and would enable investigators to obtain results that are suitable for both previous and future studies.

From the perspective of an increasingly sustainable and well thought-out agriculture, molecular tools play a pivotal and essential role; genetics and genomics, in particular, are part of a breakthrough that is able to facilitate several aspects of plant breeding and research. This review takes stock of our genomic understanding of the main agronomic traits of important forage species, such as those belonging to the genus *Lolium*, and points out that an increased knowledge of the regulation of these traits is fundamental in marker-based genomic breeding and the transgenic modification of ryegrasses.

**Author Contributions:** Conceptualization, G.B. and E.P.; methodology and bibliography resources, E.P.; writing—original draft preparation, E.P.; writing—review and editing, G.B. and E.P.; supervision, G.B.; project administration, G.B.; funding acquisition, G.B. All authors have read and agreed to the published version of the manuscript.

**Funding:** This study was carried out within the research contract signed with the seed company Mediterranea Sementi S.r.l., The corresponding author declares no conflict of interest with the seed company.

**Acknowledgments:** The authors would like to thank the seed company Mediterranea Sementi S.r.l. for funding the PhD project of Elisa Pasquali.

**Conflicts of Interest:** The authors declare no conflict of interest.

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
