# Peer review of "Genomics Applied to the Analysis of Flowering Time, Abiotic Stress Tolerance and Disease Resistance: A Review of What We Have Learned in Lolium spp."

_agriculture, doi:10.3390/agriculture10100425_

Round 1

Reviewer 1 Report

The manuscript is nice contribution to the field of grass genetics and breeding. Minor changes are required: 

L11-12: Change to 'Understanding the genetic control of these traits is enabled by the combination of genomic toools with conventional breeding techniques.'

L30: replace 'two main species' to 'two economic species'

L31: delete 'northwestern'. Ryegrasses are widely used across all Europe. 

L32: add 'New Zealand'

L34: replace 'grasslands' to 'lawns'

L37: provide citation

L40: replace 'levels' to 'production'

L40: use adequate genome size (see https://www.forages-eucarpia.org/perennial-ryegrass/)

L40-41: not all ryegrasses are cross-pollinators. Modify the text acoordingly. 

L43: use the more proper citation, such as Torrecilla and Cataln, 2002 or K. Polok 2007 (Molecular evolution of the genus Lolium L.)

L60-61: you describe that: Indeed, conventional breeding of forage grasses is mainly based on recurrent phenotypic selection aimed at the production of synthetic populations using intercrosses of a limited number of parental plants [4]. Therefore, Lolium spp. cultivars are usually characterized by a high level of within-cultivar diversity.'  I do not see the logical link between limited number of parental plants on one side and High level of diversity on the other. Please modify the text accordingly. 

L65: Citation of the review papers (such as Kopecky and Studer, 2014) on molecular methods and techniques for grasses would be beneficial. 

L67 and elsewhere: replace 'varieties' to 'cultivars'. Variety is taxonomic term.

L107: replace 'mainland' to 'Continental'

L120: replace 'a syntenic association' to 'synteny'

L132: replace 'As is' to 'As it is'

L134: replace 'and autonomous' to 'and autonomous pathway'

L182: replace 'OsHd1' to 'OsHD1'

L229: add reference

L234: add reference

L235: replace 'to create' to 'to develop'

L235: delete 'of and resistant'

L247 and elsewhere (especially page 9): replace 'resistance' to 'tolerance', once speaking about abiotic stresses

L273: replace 'action' to 'acting'

L282-299: Several interesting studies, especially those involving hybrids, are missing, such as Humphreys et al., 2005 (TAG), Kosmala et al. 2006 (Heredity) and Baird et al., 2014 (Crop Sci). The frost tolerance study of Bartos et al. (2011) in TAG is also missing.

L302: replace 'homoeologs' to 'orthologs'

327: replace 'Tauschii' to 'tauschii'

L370: replace 'considered potential' to 'considered as potential'

L399: replace 'turf' to 'forage'. Italian ryegrass is not used for turfs and amenity lawns. 

L405: add reference

L409: delete 'As very important genes and to facilitate their inheritance,'

L411: modify the reference to number

L413: modify 'Studer' to 'Studer et al.'

L414: modify 'on LG4 a major QTL' to 'a major QTL on LG4'

L425-426: delete 'biologically interesting'

L433-435: The sentence is not logically structured. Please modify it. 

L481 and later: This is the problematic part: we do not say 'on the top region of the LG'. There is frequently no knowledge where is a centromere and thus, the orientation of the LG is frequently wrong compare to physical length. If you know where is the short arm and where is the long arm and the orientation is verified (by cytogenetic studies), you can say: 'in the distal part of the short arm'. This is the right term.

L495_ add reference

Tables 3+4 - merge them and add the results of Fujimori et al.

L510: modify 'studies' to 'study'

L519, L529: see comments for L481

L535 and elsewhere: you overuse the word 'Indeed' (15x)

L546: add reference

L556 and L560: see comments for L481

Author Response

Reviewer 1

L11-12: Change to 'Understanding the genetic control of these traits is enabled by the combination of genomic tools with conventional breeding techniques.'

Correction done.

L30: replace 'two main species' to 'two economic species'

Fixed.

L31: delete 'northwestern'. Ryegrasses are widely used across all Europe. 

Thank you for the clarification. Correction done.

L32: add 'New Zealand'

Already present in line 31.

L34: replace 'grasslands' to 'lawns'

Done.

L37: provide citation

Thank you, we added the work of Reheul et al. (2010).

L40: replace 'levels' to 'production'

Replaced.

L40: use adequate genome size (see https://www.forages-eucarpia.org/perennial-ryegrass/)

We edited with the precise genome size for both L. perenne and L. multiflorum.

L40-41: not all ryegrasses are cross-pollinators. Modify the text acoordingly. 

Thank you for the clarification, we edited specifying the different species providing also the reference.

L43: use the more proper citation, such as Torrecilla and Cataln, 2002 or K. Polok 2007 (Molecular evolution of the genus Lolium L.)

Thank you for the suggestion, we added them.

L60-61: you describe that: Indeed, conventional breeding of forage grasses is mainly based on recurrent phenotypic selection aimed at the production of synthetic populations using intercrosses of a limited number of parental plants [4]. Therefore, Lolium spp. cultivars are usually characterized by a high level of within-cultivar diversity.'  I do not see the logical link between limited number of parental plants on one side and High level of diversity on the other. Please modify the text accordingly. 

Sorry for this unclear part of the manuscript. Following your suggestion, we edited it as follows “Conventional breeding of forage grasses is mainly based on recurrent phenotypic selection aimed at the development of synthetic populations using intercrosses or polycrosses of a limited number of parental plants selected for their general combining ability. Nevertheless, the outcrossing reproductive system of Lolium spp. coupled with its strong self-incompatibility system leads to obtain cultivars usually characterized by a high level of within-population genetic diversity.”

L65: Citation of the review papers (such as Kopecky and Studer, 2014) on molecular methods and techniques for grasses would be beneficial. 

We added it together with another reference (Harris-Shultz et al., 2018).

L67 and elsewhere: replace 'varieties' to 'cultivars'. Variety is taxonomic term.

Thank you for the clarification, we edited it throughout the text.

L107 (now 111): replace 'mainland' to 'Continental'

Done.

L120 (now 125): replace 'a syntenic association' to 'synteny'

Done.

L132 (now 137): replace 'As is' to 'As it is'

Done.

L134 (now 139): replace 'and autonomous' to 'and autonomous pathway'

Done.

L182 (now 188): replace 'OsHd1' to 'OsHD1'

Done.

L229 (now 235): add reference

We added the studies of Bellard et al. (2012) and Ergon et al. (2018).

L234 (now 239): add reference

We added references for different metabolic response (Acosta-Motos, 2017; Chalanika, 2017 and Heidarvand, 2010).

L235: replace 'to create' to 'to develop'

Replaced.

L235: delete 'of and resistant'

Deleted.

L247 and elsewhere (especially page 9): replace 'resistance' to 'tolerance', once speaking about abiotic stresses

Thank you for the tip, we replaced it throughout the text.

L273: replace 'action' to 'acting'

Done.

L282-299: Several interesting studies, especially those involving hybrids, are missing, such as Humphreys et al., 2005 (TAG), Kosmala et al. 2006 (Heredity) and Baird et al., 2014 (Crop Sci). The frost tolerance study of Bartos et al. (2011) in TAG is also missing.

Thank you for these suggestions, we added them with this reference “In fact, introgression of F. pratensis genes in Lolium genomes has often represented a great way to improve also freezing and drought tolerance developing hybrid plants [Kosmala, 2006; Humphreys, 2005; Bartosˇ, 2010].

L302: replace 'homoeologs' to 'orthologs'

Done.

327: replace 'Tauschii' to 'tauschii'

Done.

L370: replace 'considered potential' to 'considered as potential'

Done.

L399: replace 'turf' to 'forage'. Italian ryegrass is not used for turfs and amenity lawns. 

Thank for this clarification, modified.

L405 (now 415): add reference

We added the references of Schmidt et al. (1980) and Wang et al. (1995).

L409: delete 'As very important genes and to facilitate their inheritance,'

Deleted.

L411: modify the reference to number

Modified this one and others throughout the text.

L413: modify 'Studer' to 'Studer et al.'

Done.

L414: modify 'on LG4 a major QTL' to 'a major QTL on LG4'

Done.

L425-426: delete 'biologically interesting'

Deleted.

L433-435 (now 443-446): The sentence is not logically structured. Please modify it. 

This sentence was actually unclear and hence it was edited and rewritten as follows: “Another candidate gene for bacterial wilt resistance detected in L. multiflorum by Knorst [108] encodes a noduline MtN3 family protein (also termed SWEET17 in Arabidopsis). Interestingly, its homolog in rice, SWEET13, was demonstrated to encode a protein (known as xa25) which confers resistance against X. oryzae pv. orizae [110].”

L481 (now 492) and later: This is the problematic part: we do not say 'on the top region of the LG'. There is frequently no knowledge where is a centromere and thus, the orientation of the LG is frequently wrong compare to physical length. If you know where is the short arm and where is the long arm and the orientation is verified (by cytogenetic studies), you can say: 'in the distal part of the short arm'. This is the right term.

Thank you for this important clarification. We apologize for the mistake. In this case we changed as suggested because Roderick et al. (2003) provided a physical map of the genes identifying the region with a genomic in situ hybridisation (GISH) analysis.

L495 (now 506-507) add reference

We added references of the works of Jones et al. (2002) and Yu et al. (2000). 

Tables 3+4 - merge them and add the results of Fujimori et al.

We thank you for the suggestion but after careful deliberation we agreed to keep table 3 and 4 separated because the information derived from independent experiments whose results can not be overlapped.  

L510: modify 'studies' to 'study'

Done.

L519, L529: see comments for L481

In these cases, we delated the information of the position since it does not derive from a physical map.

L535 and elsewhere: you overuse the word 'Indeed' (15x)

We edited this aspect throughout the text.

L546 (now 560): add reference

We added the references of the works of Feuillet et al. (1997) and Cheng et al. (2002). 

L556 and L560: see comments for L481

In these cases, we delated the information of the position since it does not derive from a physical map.

Reviewer 2 Report

This most parts of information in this MS did not well center around Lolium spp. And the way authors described in QTL part sounds very forced.

Some suggestions are shown below: 1, Line 113 to 120 and Table 1: Since different types of markers and populations were used in these studies (Table 1) and lack of reference genome and common markers during the time when experiments were taken, the correlations between LGs across that authors described is very forced and may be not valid, unless authors provides more evidences to correlate these studies.

2, Line 132 to 224: Those parts are unnecessarily long. Authors should primarily focus on the crop of interest but abbreviate the general knowledge.

3, Line 227 to 395: Please refer to question #2.

4, Line 416 to 417: This conclusion inferred by comparing QTL effects between plants sounds very forced.

5, Line 423: what are “innovative genetic markers”?

6, Line 423 to 446: Authors described findings in other plant species (some are relatives and some are not) and wrote that L. multiflorum possessed the similar kind of mechanism (line 441 to 443). This part can be incorporated in every case, yet failed to provide useful information that was specific to Lolium spp.

7, Line 493 to 495: Lack reference here.

8, Line 482 to 579: Please refer to question #1. Otherwise, the information described here is not useful.

9, Latin names were not properly italicized. Authors should check throughout the MS

10, Line 581 to 620: Please refer to question #1 and 8

Author Response

Reviewer 2

1, Line 113 to 120 and Table 1: Since different types of markers and populations were used in these studies (Table 1) and lack of reference genome and common markers during the time when experiments were taken, the correlations between LGs across that authors described is very forced and may be not valid, unless authors provides more evidences to correlate these studies.

Thank you for your comment, we edited this part stressing the limits of the available data and information as follows “It is essential to underline that, since they were identified in different studies, the correlation between LGs and the identity of the loci involved can hardly be determined due to the distinct mapping populations, the lack of a reference genome, trial locations used and the scarcity of common molecular markers. Nevertheless, the results of these studies confirm the strong genotype-by-environment interaction that always has to be considered in studying heading date.”

2, Line 132 to 224: Those parts are unnecessarily long. Authors should primarily focus on the crop of interest but abbreviate the general knowledge.

As suggested by your comment, we shortened the general knowledge section by removing unnecessary information for the understanding of mechanisms described later throughout the text, such as those on lines 139-144 and 181-187.

3, Line 227 to 395: Please refer to question #2.

Here too, we removed some general redundant parts that can result not so fundamental as lines 245-252, 337-342 and 396-404.

4, Line 416 to 417 (now 427-429): This conclusion inferred by comparing QTL effects between plants sounds very forced.

Thank you for your comment. In this case, the information about the relationship between Lolium LG4 and rice Chromosome 11 was inferred from the referenced papers by Studer et al. (2006) and Devos (2005). Moreover, we referred to the collinearity of the two genomic regions.

5, Line 423 (now 433) what are “innovative genetic markers”?

We edited the text in this way “Since the use of molecular markers in breeding programs is increasingly common, a set of 21 SNPs…”

6, Line 423 to 446: Authors described findings in other plant species (some are relatives and some are not) and wrote that L. multiflorum possessed the similar kind of mechanism (line 441 to 443). This part can be incorporated in every case, yet failed to provide useful information that was specific to Lolium spp.

7, Line 493 to 495 (now 506-507): Lack reference here.

We added references of the works of Jones et al. (2002) and Yu et al. (2000). 

8, Line 482 to 579: Please refer to question #1. Otherwise, the information described here is not useful.

Here too, we remarked the limits of referring to different independent studies adding the following sentence on line 526-528 “Although the clear difficulty to define a correlation between the QTLs detected in absence of a reference genome and common markers during the time when experiments were taken, it is important to point out the strong interaction between resistance QTLs…”. Moreover we removed some redundant parts as those on line  509-511, 516-518, 523-526 and 542-544.

9, Latin names were not properly italicized. Authors should check throughout the MS

Checked throughout the text and fixed. 

10, Line 581 to 620 (now 597-636): Please refer to question #1 and 8

Here too, we specified the different origin of each experimental result and deleted forced and superfluous information such as those on lines 612-613 and 629-636.